# Creating Sustainable Order Fulfillment Processes through Managing the Risk: Evidence from the Disposable Products Industry

**Mohammad Heydari [1]** **, Kin Keung Lai [2],\* and Xiaohu Zhou [1]**

[1]  School of Economics and Management, Nanjing University of Science and Technology,
    Nanjing 210014, Jiangsu, China; Mohammad_Heydari@njust.edu.cn (M.H.); zxhnjust@njust.edu.cn (X.Z.)
[2]  College of Economics, Shenzhen University, Shenzhen 518060, China
\*  Correspondence: mskklai@outlook.com; Tel.: +86-852-2859-2586

**Abstract:** Retailers face a major operational challenge in fulfilling online orders while managing their traditional store-based distribution processes. In this context, the following order fulfillment options available to retailers are considered: store-facing distribution centers (DCs), dedicated order fulfillment facilities (DTCs), retail stores, and direct-fill by vendors. The current study provides an order fulfillment evaluation for the Disposable Products Industry, which is one of the industries that have a tremendous effect over the downstream industries, as it is the source for production. Also, the differences in the factor focus are provided for various parties and countries. The results show that the order fulfillment risk factors identified from various research studies are good enough for the Disposable Products Industry, even though they are not intentionally designed for this highly diversified industry. Among them, sustainability is the most important factor that the companies in the Disposable Products Industry should pay attention to. This is because sustainability is believed to lead to large deviations in various types of order fulfillment losses and incur a higher chance of having the order fulfillment failure for the companies with an international customer base. The companies should focus on how to improve the sustainability (long-term relationships with the various parties along the chain) rather than over-emphasis on the short-term documentation accuracy as the long-term improvement is likely to result in an overall improvement in performance on order fulfillment.

**Keywords:** risk management; fulfillment; disposable products industry; sustainability; supply chain

## 1. Introduction

For the past decade, online retail sales have increased at a faster rate than sales through retail stores. In 2015, online sales in the United States were estimated at USD 341 billion (an increase of 14.6% over the previous year), whereas store sales during the same period increased by only 1.4% [1]. The growth of online sales has attracted many "pure-play" brick-and-mortar store retailers to add the online channel to their business strategy. These retailers seek to align their traditional store-based distribution processes with the requirements of the online channel through coordinating demand management and order fulfillment activities [2]. This realignment of supply chain processes has resulted in the emergence of a retail distribution strategy in which retailers organize unified order fulfillment processes that utilize different nodes in the distribution network to fill customers' online orders. The fulfillment nodes include retail stores, distribution centers (DCs), order fulfillment facilities and vendors [3].

While adding the online channel offers retailers access to new customers and better sales prospects, executing the online order fulfillment process within a store-based distribution network creates

significant operational issues [4]. For example, consider the differences in the profiles of online orders and store replenishment orders. Online customers buy in small quantities per order with different delivery (next-day, three-day) and destination (direct to customer homes or pick-up from local stores) requirements. On the other hand, store replenishment orders are larger and require picking and shipping full-case pallet loads. Filling the respective types of orders requires different order fulfillment activities and arrangements of resources. Processing high volumes of small-sized online orders requires pick-pack-ship activities that are either labor-intensive (and thus costly) or mechanized (which requires significant capital investment).

A key element in the online order fulfillment process is the last-mile delivery of orders. Store-based retailers are not used to providing this service. For decades, these retailers have successfully handled product flows from their DCs to the stores; however, delivery of orders to customers' homes is something new for them, which they have been struggling to manage [5]. The challenge posed by differences in order fulfillment needs and last-mile delivery has caused serious downward pressure on store-based retailers' financial bottom lines [6]. Our article focuses on the abovementioned issues related to order fulfillment and delivery and presents a study of the order fulfillment process used by retailers engaged with the store and online channels.

Order fulfillment analysis has long been focused on the downstream supply of products (finished products). As stated by the International Disposable Products Federation, the failure of the facilities of an India petrochemical supplier can cause serious disruption to the downstream Disposable Products supply. It is worth noting that the supply of materials from the very upstream end, like the Disposable Products raw materials or semi-finished products, can lead to tremendous effects on various productions across the industry.

As shown in Appendix A, the usage of Disposable Products can be very diversified. Whether for sole support functions like packaging, to industrial usage like automotive, up to general consumer product production, the use of Disposable Products is inevitable.

Although many managers consider order fulfillment to fall within the role of the logistics function, it is the integration with other functions in the firm and other firms in the supply chain that becomes key in defining order fulfillment as a supply chain process.

The usage diversification of Disposable Products is not country-specific and unique. According to the National Bureau of Statistics of the People's Republic of China (2015a,b), the proportion of the rubber and Disposable Products finished products within the overall manufacturing of finished products (in billion RMB) is ever increasing and the value is over RMB 100 billion per year [7,8].

With the importance of order fulfillment to the operation (supply chain) and the significance of the Disposable Products Industry, the current research would like to provide an analysis of the topic: Risk Management in Order Fulfillment for the Disposable Products Industry in China. The research would like to find out: "What are the risk factors and how would they hinder the order fulfillment process of the Disposable Products Industry in China? Are there any references to better the process?"

The scope of the analysis is the operations and business relationship of multinational firms that provide the Disposable Products material to firms in China for any of the production. The objectives of the research study are as follows:

- Finding out the factors that affect the order fulfillment process for the Disposable Products Industry in China,
- Contrasting the factors that affect the order fulfillment process for the Disposable Products Industry in China and other countries,
- Identifying the differences in focus for the order fulfillment process among various parties in the Disposable Products Industry,
- Investigating the relationship of the focus for the order fulfillment process and the performance of order fulfillment for various parties along the chain in the Disposable Products Industry,
- Identifying the normal practices for the order fulfillment process that are used by the companies with outstanding order fulfillment performances.

We developed a modeling framework for the retail distribution system in which online orders can be filled from different fulfillment nodes. The framework applies existing constructs of network flows in the operations research literature to a new study focused on the order fulfillment process. This framework incorporates different order fulfillment paths and implements the decision logic used by retailers to select an appropriate mix of order fulfillment options. The selection of fulfillment options is based on the intrinsic operational attributes: product flows, logistics, and operational costs, and storage limits related to every fulfillment node in the network. We used this model to conduct an empirical evaluation of the order fulfillment process using real-world data collected from a large U.S. retail firm by considering various configurations within an experimental design study. Results were analyzed to develop managerial insights regarding operational factors that affect fulfillment choices in a retail supply chain.

## 2. Related Research

### 2.1. The Order Fulfillment Process and its Relationship with Other Business Processes

Creating value is always the focus of a company and value is believed to be created through the order fulfillment process. (See Appendix A, Figure A1.) Is a practical order fulfillment value stream process one that shows how the value is created along with the flow of the process? As the figure shows, production is a part of the whole value stream. Order fulfillment, on the other hand, overlies the whole customer satisfaction process [9]. It goes from receiving the order to producing the required order, cashing the invoice and after-sales support.

Even though the operational order fulfillment process is listed separately from the other supply chain processes and is believed to be conducted within the logistics function, it is not separable from the other business processes. Therefore, to understand the order fulfillment it is necessary to focus on the interaction with the whole supply chain system [10]. Appendix B, Figure A2 demonstrates the detailed relationship between the transactional order fulfillment processes and the other supply chain process. The transactional order fulfillment is very similar to [9]. As proposed by the Global Supply Chain Forum, order fulfillment is one of the eight central processes for the whole supply chain management process (seven other processes are listed on the left of 0, see Appendix A, Figure A2).

Order fulfillment is believed to be a complex process because it involves several processes and activities that are carried out by different involved entities or functional units. There is a strong linkage between the duties, resources, and agents to complete these activities. As stated by Lin and Shaw (1998), many of the manufacturing processes are outsourced, and thus the movement of semi-finished products involves even more entities, and the upstream and downstream link cooperation becomes even more engaged. It leads to the importance of having good planning in terms of supply chain concern [11]. As shown in Appendix A, Figure A3, order fulfillment is as important as the product development and customer service, which are regarded as the core business processes of the company and are carried out through the coordination and collaboration of all the functional activities within a firm (at the bottom) and the entities outside the firm to produce a final product to the customers at the end of the process [12].

If the focus is extended systematically, with the incorporation of the supply chain management in mind, order fulfillment still plays an important role in the whole process. The complex relationship does not only rely on the transactional relationship development as shown in Appendix A, Figure A2; it is shown in Appendix A, Figure A4, that the strategic planning process is interactive with the seven important supply chain processes as identified by the forum. The requirement and services differentiation is affected by customer demand and the consideration of the production capacities [13]. As there would be different capabilities and characteristics of the production plant, transport route, warehouse and distributors' locations or others, the actual plan for order fulfillment would differ, taking into account the supply chain elements and structures. The actual performance is highly effective in customer relationships and further marketing strategies. Therefore, Stock and Lambert

(2001) suggested that order fulfillment is highly related to marketing, customer services, and supply chain network. Customers should be satisfied by the sub-products of order fulfillment [14].

Other than the smooth operation, order fulfillment is believed to influence loyalty. Unless otherwise engaged, downstream parties like the retailer can shift from one manufacturer to another. Mitchell (2004) stated that manufacturers compete for retail business as virtually faceless vendors; [15]. Davis-Sramek, Stank and Mentzer (2008; 2010) showed that to make the customer stay with the business [16,17], an understanding of order fulfillment service capabilities is needed. The ability to turn faceless vendors to value-added partners depends on the ability of the firm to provide service in the order fulfillment process that can affect the customer's (retailer's) perception. Fuller, O'Connor and Rawlinson (1993) suggested that price and quality sometimes can provide minimum expectations for manufacturers [18]. On the other hand, the ability to provide highly appreciable logistics and order fulfillment processes can make the company more competitive.

Davis-Sramek, Germain and Stank (2010) confirmed that order fulfillment can influence the retailer perceptions [17]. The order fulfillment process is believed to go beyond the technical aspect of the logistics process for shipping the goods on time to maintaining some relationship with the retailers, like the ability to handle dissatisfied customers, reliability, and timeliness. Relationship maintenance can be built through the successful order fulfillment obligation rather than the functional task (order delivery) impact on efficient flow only. When going one step further, the order fulfillment process can be used to investigate the repurchase intention with their past order fulfillment experience. This impact is important because such an intention is highly influential to the future revenue of the company [19].

Zhang, Jiao and Ma (2010) said that the importance of the supplier in the order fulfillment process is often ignored with the overemphasis on the final product producers when the supply chain network objective is defined as satisfying the customer requirement [20]. Therefore, the research suggested that the reengineering of the order fulfillment process is crucial. The use of simulation techniques on the order fulfillment process presents a significant improvement in the whole supply chain network. Practically, according to the metrics developed to measure the performance of the supply chain network (Stephens, 2001), the supply chain can be evaluated through various attributes [21]. Among these, order fulfillment is highly related to the delivery reliability and responsiveness of the supply chain. The former is related to the correctness of the delivery, and the latter is about the speed and time of the supply chain movement. In other words, order fulfillment is crucial in maintaining the effectiveness and efficiency of the whole supply chain network.

## 2.2. Variation in Concerns and Strategies over Order Fulfillment

When talking about actual order fulfillment practices, many of the research studies focus on the smooth operation of the process. One of the major issues is the order picking decision and strategy determination. Petersen and Aase (2004) suggested that order picking is the major cost component of order fulfillment [22]. Therefore, the authors investigated the picking, routing and storage policy that would affect the order picking operation. With policies like batch picking, class-based storage, and optimal routing, the operation time and fulfillment can significantly improve with less picking time, and an increase in picking size. De Koster, Le-Duc and Roodbergen (2007) said that order picking not only affects the operation of the warehouse but the whole supply chain movement [23]. Other than picking, routing, and storage, the layout of the warehouse and zoning were also analyzed. Also, it was suggested that daily and rolling demand should be incorporated in the picking process so that better strategies can be provided.

On the other hand, it is also obvious that physically delivering the goods to the customers would affect the order fulfillment, according to scholars like Bullinger, Kuhner and Van Hoof (2002) and Lapide (2000) [24,25]. They showed that order fulfillment is highly related to some transportation-related variables like the number and cost of transportation and storage facilities, the ability to track the procedures with accurate invoice orders, the number of backorders and delayed orders and the flexibility of transporting the goods. When talking about the transportation time, Lin and Shaw (1998)

divided the time element into the reception and inspection time, assembly release time, material order picking time, transport time through stages, transportation time to customers and final assembly time on the customers' side. It was found that the percentage citing the use of a third-party logistics service by the U.S. firm increased from 11% in the late 1990s to over 30% [26]; this may be due to the importance of logistics functions over better order fulfillment engagement.

### 2.3. Special Issues over Order Fulfillment

Section 2.2 summarized some of the key concerns, factors, and arrangements regarding various types of order fulfillment. It includes the order picking, logistics, and transport arrangements, product concerns, and supply chain corporations. These explain why there are numerous studies that can lead to different strategies when tackling the same problem. Just like many other research areas, the study of the order fulfillment is ever-changing with different issues to be addressed over time. Here, two important and highly concerning issues—information sharing for order fulfillment issues with information technology concerns and sustainable order fulfillment—are discussed to provide even more inspiration to the current research proposal.

Cotteleer and Bendoly (2006) showed that after the implantation of the enterprise resources planning system (ERP), there were improvements in the performance, like order lead time and change in inventory level [27]. Customer orders can be fulfilled with a decrease in inventory buffer. The order and inventory handling process can be more standardized, which reduces the operational variability [28]. The order fulfillment IT system does not only manage the order but also standardizes and integrates the understanding of the process from the planning, purchasing and production [29]. To be more exact, it includes the order management (order entry system, order automation, payment transfer), resources planning (SKU management, facility design, warehouse management), production scheduling (master, weekly and daily), capacity planning, material planning (inventory planning, replenishment, product flow), production control (supply chain management), packaging and shipping (shipment consolidation, tracking, notice, routing) and others (profiling, sharing of practices and strategic information, supplier monitoring) [11,30]. More importantly, it was suggested that IT-driven continuous improvement efforts are opportunities for overall operational efficiency enhancement. Continuous improvement in IT is foreseeable because of the change in organizational learning after the system implementation. The digitalized process is an IT-enabled inter- and intra-organizational process that makes the activities atomized, informative and integrated [31]. In this way, information can be shared seamlessly. There is no limitation on the infrastructure. Order information can pass through the system from the downstream entities like distributors to first-tier or beyond-the-first-tier suppliers such as the upstream receivers [11]. This enables the proposition by Hult, Ketchen and Nichols (2002), who believed that information technology is important to order fulfillment because it can facilitate the sharing of common beliefs and values [32]. They are more willing to cooperate to finish the tasks with such common shared competitiveness. The order fulfillment process in a dynamic supply chain network can be improved through the improved information infrastructure, which can be shown from the market response, decrease cycle time and stable inventory cost [33]. In the long run, companies can come up with re-engineering considerations through the use of information technology to align with the business process design as it is accompanied by the possibility of continuous improvement and streamlining of the daily operation [34]. Within this re-engineering process, the customer-driven business processes like order fulfillment are being highlighted to integrate with the IT capabilities of the company [35]. By enhancing the speed and accuracy of the order processing through the demand planning of the IT-enabled order management system, superior experiences can be produced to the customers and business partners [36].

Another important issue that has come to attention is the sustainability of the order fulfillment. Putri et al. (2013) revealed that demand fulfillment needs to cooperate with the consideration of the harvest and conserve the scarce resources [37]. This is highly related to the benefit of people all over the world. The various dynamic concerns are incorporated, which makes the balance between harvest and

conserve very complex, for example, the classes of the resources, demand, regenerating of the resources, and illegal production. However, the definition of sustainability may not be limited to the ability to adjust the environmental regulations for the protection of the future. It can be very broad, in which the needs of the order fulfillment can be achieved at present without compromising the future benefit. Troyer et al. (2005) invested in the enterprise planning and execution systems because they expect a sustainable inventory reduction [38]. This is a difficult task because there is a need to balance supply and demand across various levels of production and storage, inventory type, multiple locations in a continuous manner. To achieve a longer and more sustainable improvement, both the problems (e.g., market change, customer linkage, channel bond) and the benefits associated with the order fulfillment management should be considered together so that it can lead to the identification of the capability of critical elements rather a simple activity sequence identification [39]. Other than the multiple functions and processes that need to be addressed together, Miemczyk, Johnsen and Macquet (2012) suggested that there is often a lack of analyses of the integration of the supplier in the order fulfillment process via a planning and execution system [40]. It is believed to be one of the possible factors that affect the sustainability of the order fulfillment system. Sustainability needs to be extended outside the firm alone. Lee (2004) suggested that companies are focusing on the ability to develop a demand-driven supply network in which technologies and business processes can provide a response, and the real-time demand from the network of supply chain parties can attain a sustainable competitive advantage [19]. Practically, Hajdul (2010) showed that by coordinating the transport process with the incorporation of the social (e.g., congestion and average speed of the road), technology and environment concerns (e.g., emission), the transport cost, distance, number of individual transports, the order fulfillment time could be improved [41]. One of the objectives and practical, sustainable supply chain management perspectives is that it can integrate the raw materials procurement production handling and material distribution [42]. More importantly, the system is sustainable because it can only accompany a process that is effective, like the high concern over the demand forecasting and the performance on delivery. For example, only the company that can meet over 80% of the order fulfillment rate should remain because the product quality needs to be maintained to provide consistently high standard products to the downstream.

## 3. Conceptual Modeling: Definition, Purpose, and Benefits

Previous research did not clearly define the risk factors in order fulfillment for an industry that has such broad usage. Usually, the research studies focused on one or two end products (downstream) of the industry for comparison and analysis. However, an industry that has broad usage is more important as its impact towards various end products is far beyond what one can imagine.

No research has tried to contrast the different focuses on various risk factors for order fulfillment among different parties along the supply chain. With the different natures and target customers and partners, their focuses may vary a lot.

Research studies have focused mainly on the cases on the western side of the world. It is important to look at the case from another side of the world. Would the risk factors for order fulfillment being considered by the various parts of China be treated in the same manner?

There is hardly a research study evaluating the order fulfillment that tries to contrast the actual concern of the company and the practices and focuses that may be important for perfect performance (which can be found by analysis of the risk focus of the relatively high order fulfillment performance). It is important for a company to evaluate its own performance and identify the deficiencies that may lead to relatively low order fulfillment performance.

A possible solution for improving order fulfillment can be found. However, this cannot be done without identifying the cause of the disruption. The studies are diverse (without the consideration of various background characteristics) and do not address the issues with the risk identification. It is more important to find out the most influential risk factors in the order fulfillment process when there are different priority order fulfillment measurements. Many studies propose that the most useful way

to solve the disruption in order fulfillment is through information sharing, but is it the only way to end problems?

*3.1. Risk Factors for Order Fulfillment*

The first part of the concept map is to find out factors that concern the Disposable Products Industry in China. These factors are identified by various research studies that are believed to be influential to the order fulfillment process (see Appendix A, Figure A5).

"What are the risk factors involved in the order fulfillment process for the Disposable Products Industry in China?"

Lapide (2000) believed that a logistics process would provide the effectiveness to the order fulfillment, which includes the routing [25], tracking ability, invoicing accuracy and backorder fulfillment. Lin and Shaw (1998) further divided the time element concerned by the logistics process [11]. This is accompanied by the order picking process as suggested by Agatz, Fleischmann and Van Nunnen (2008), who believed that the fulfillment process is complex in the sense that it involves various processes from sourcing [43], order picking in the warehouse, actual delivery and the sales (reaching the customers).

Pil and Holweg (2004) thought that order fulfillment is influenced by the level of customization [44], in order words, the variation by customers and internal process. However, in some situations, even when the demand is present, with the regulations, the supply and order fulfillment may be constrained [45].

When linking up the various parties in this multi-channel and multi-party network, integration and relationships among the various parties are important. Forslund and Jonsson (2010) believed that order fulfillment could be affected by the different levels of integration in the performance management, which include the fulfillment metrics design, target setting, measurement, and further analysis and control [46]. To some extent, this concerns the order fulfillment in the long run or the sustainability of the process. The social and environmental concepts may come to mind at first sight [41,47] and concern over regulation is important. Also, Lee (2004) suggested that the ability to develop a good relationship with the other parties, like the ability to provide a demand-driven supply network, would lead to a competitive advantage [19].

Last but not least, the total supply chain operation is crucial when the firm is operating in the available-to-promise mode or the make-to-order way, which would make a difference in the order fulfillment strategies [48,49].

However, the research studies we have shown above are not coming directly from the Disposable Products industry, which is a special industry that has a very long supply chain that involves parties from the very upstream, like material sourcing, and the very downstream that include the industrial and consumer products. The mixture and broadness of products make the order fulfillment risk factors vary. Therefore, some more ideas for risk factors in the industry are needed. Thus, an in-depth interview would supplement the factors identified in the previous researches by asking for the general risk factors in the Disposable Products Industry. This would include requests like:

- Identify some risk factors that are involved in the Disposable Products Industry,
- State their frequencies and the impacts,
- Identify the role and significance of the order fulfillment process of these risk factors,
- Name the specific risk factors involved in the order fulfillment process,
- State the internal and external perspectives,
- Name the departments (internal) and parties (external) involved in the order fulfillment process with the company,
- Describe the relationship with these parties and departments and their roles,
- State the frequencies and the impacts of these internal factors and external factors.

This part will identify the risk factors that are influential to the Disposable Products industry, which is useful for the sequence analysis and solves the problems below. Moreover, the respondents would be asked to evaluate the importance of these factors to the company itself.

*3.2. Order Fulfillment Risk Factors Variations for Different Characteristics*

As shown from the literature review (Section 2.2), the order fulfillment process focuses are different for different industries, cultures, and characteristics. Therefore, it is important to look at how the risk factor focus varies among different roles along the supply chain, product types, and culture.

"Are there any differences in risk factors considered for various characteristics in the Disposable Products Industry?"

The above concept map is also used to contrast the factor concerns among the Chinese firms and the other countries' corporations. This can be done as some of the companies being surveyed (mainly those at the upstream parts of the supply chain) would be multinational countries (see Appendix A, Figure A6).

A similar comparison would be provided to identify the differences in concern among various supply chain parties, as their concerns may differ a lot with different operation environments (see Appendix A, Figure A7).

The concept maps demonstrate the factors that are believed to be risks related to the order fulfillment process in a general situation. In addition to comparing these underlying risk factors among the Chinese firms and other multinational firms, as well as contrasting the various roles along the supply chain, the in-depth interview would be used to ask for the possible factors that can be added to the conceptual map for large-scale risk factor analysis across countries and layers.

With this concept in mind, the respondents in the in-depth interview would be asked to consider among the factors that have been stated, which of them are specific to a particular:

- Industry
- Product
- Firm
- Role
- Country

This confirmation of factors in a specific context (industry, product, firm, etc.) can help to set up the detailed hypotheses for contrasting the factor importance in different contexts for the survey questionnaire in the later sections (which will be set up after the in-depth interview). This is important as it can modify the factors in previous studies (which are not Disposable Products industry-specific).

As the first two parts of the study are to evaluate the impact of various risk factors over the order fulfillment process, this may not show the complete picture of the story. To increase the soundness of the results and be more objective, measurements of the order fulfillment process are needed.

"What are the measures that would be used for evaluating the order fulfillment process?"

"Are there any relationships between the risk factors consideration and the performance in the Disposable Products Industry in China?"

"Are there any perfect order fulfillment guidelines and norms in the Disposable Products Industry in China?"

The next part of the concept map as shown in Appendix A, Figure A7, evaluates the effect of various order fulfillment factors on the performance of order fulfillment. Other than the perfect order fulfillment performance criteria as suggested by Mishra and Sharama in 2014 [50], the time of the order fulfillment is further divided by Song, Xu and Liu (1999) into response time and waiting time [51]. Hult, Ketchen and Nichols (2002) also suggest that the performance can be evaluated subjectively [52]. This includes but is not limited to the assessment of the satisfaction with the order fulfillment and whether the time performance can be shorter than it is at the moment.

Also, the respondents in the in-depth interview would be asked to:

- State the measures and the task forces for performance measurement over the order fulfillment process,
- State the regulations and norms provided by the industry and regulatory bodies,
- State whether the process measurements are specific to:

  ○　　industry
  ○　　product
  ○　　firm
  ○　　role
  ○　　country,

- Rank the importance of the measures,
- Define the perfect order fulfillment process,
- State the relationship between the order fulfillment and the various risk factors.

The respondents in the survey process would need to rate the company's performance. Then, companies with perfect and imperfect order fulfillment performances can be identified to see their ratings on various factors and whether specific focuses can help the fulfillment process.

## 4. Data Analysis

### 4.1. Qualitative Research Analysis Approach

As the Disposable Products Industry has not been structurally investigated beforehand, risk factor identification for order fulfillment would rely on the use of both deductive and inductive approaches. Deductive category development in qualitative analysis relies on the coding of the text of the in-depth interview by using the concepts provided by previous literature and studies. In this study, we would like to know that risk factors affect the operation of order fulfillment in the industry specifically. (This study would like to find out what are the factors that are relatively important in the industry.) The factors confirmed in the qualitative analysis were used for questionnaire drafting so that quantitative analysis can be carried out. In the current study, the literature would include those relying on ordering fulfillment in any industry. This includes the order picking, product-related factors, logistics, sustainability, information sharing, and supply chain strategies. The main idea is to give examples and coding rules for the coding category and determine when the text would belong to a particular category. This is important as it can confirm whether the non-Disposable-Products-related risk factors can be applied to the industry order fulfillment process.

As one may expect, some of the text cannot be grouped and coded into the pre-defined categories under the deductive category development as the reference literature is not Disposable Products Industry-oriented. Therefore, the mixing method is proposed, and inductive category development is also needed. After coding all the text with the pre-defined categories, the passage would be gone over again, and categories are tentatively defined for the un-coded text. The categories would be revised after 10%-50% of the material, and this loop is to ultimately reduce the main categories and check for reliability [53]. Axial and selective coding processes would be carried out later for further confirmation [54].

### 4.2. Quantitative Research Analysis Approach

"Are there any differences in risk factors considered for various characteristics in the Disposable Products Industry?" (This includes the company's role along the chain, the company base, the product type, and firm characteristics.)

"Are there any relationships between the risk factors consideration and the performance in the Disposable Products Industry in China?"

"Is there any perfect order fulfillment guidelines and norms in the Disposable Products Industry in China?"

All the above research questions coincide with the objectives as stated in the Introduction. The risk factors and the characteristics will be finalized when the in-depth interviews are carried out.

*4.3. Sampling and Survey Design*

Just like the in-depth interview, various parties along the Disposable Products supply chain with different bases and characteristics (firm size, year of establishment, etc.) would be included as this is the very first analysis of the Disposable Products Industry. Thus, risk factors would include ideas from various perspectives. The only concern would be the respondent's characteristics; only respondents with at least five years of experience in the industry would be included as this is to make sure that they have enough knowledge of the industry and the risk incorporated.

Initially, stratified sampling techniques would be considered. For each layer, base, and characteristic, the company would be sort-listed to form different strata. Then, samples would be selected from each stratum to include various companies in the final sample. This is to reduce the bias of the sample.

Pilot testing would be carried out first, even before the qualitative study has been carried out; this is because some of the elements in each construct (e.g., order picking) may not be consistent with each other as suggested by the interviewees of the qualitative study. Therefore, statistical testing, like the use of correlation and Cronbach's alpha, can help in checking the internal consistency of reliability and remove irrelevant elements. Face validity can be easily solved by debriefing those participants in the pilot testing. They would be asked to state whether the questions represent the pre-defined meaning or not. Also, they can help in improving the layout and wording of the questionnaire. Thirty samples would be sought for the pilot testing as this is the basic requirement for carrying out the statistical testing. Regression, t-test, and ANOVA would be the major tools for carrying out the analysis in the later stage.

Response encouragement can be done by providing the free report to the companies to provide some insights on proper order fulfillment risk factors considered. Also, continuous reminders to the respondents are needed if responses cannot be obtained.

At 0.1 level of significance, there is a significant difference in the amount arising from order fulfillment for levels from various locations. No matter whether for the level with or without significant difference, it is found that the amount that arises from order fulfillment for Hong Kong is generally lower than that of southern China. There is no significance in the amount arising from order fulfillment for levels among various entities and management. The results show that a company with an international customer base has significantly higher amounts arising from order fulfillment for all levels of loss.

Sustainability and information sharing loss have significant differences among location groups. Southern China has lower loss than eastern and northern China in the sustainability-related order fulfillment failure. For management styles, it has led to significant differences in the logistics order fulfillment failure loss. Semi-international companies have higher logistics loss than local firms.

There are significant differences in the logistics lawsuit, product nature (monetary loss, further backorder, increase in cost) and sustainability (monetary loss and further backorder) among various location groups. There are significant differences in the logistics lawsuit, order picking lawsuit, product nature-related increase in cost, sustainability (lawsuit, monetary loss, and further backorder, reputation loss and increase in cost), information sharing (lawsuit, monetary loss) and supply chain party-related increase in cost among various entity groups.

Results show that state-owned companies are more likely to suffer losses caused by sustainability than the co-operated companies.

Especially, private companies are more likely to engage in logistics lead order fulfillment related lawsuits and subsequence losses than the co-operated companies. For order picking, losses arising from the lawsuit are more likely to occur in northern China than Hong Kong/Macau, southern and

eastern China. This loss is also more likely to occur in the private-owned local company than the state-owned and co-operated firms. Also, companies with a domestic customer base are more likely to have a lawsuit due to order picking failure, while the companies with mainly international customers are more likely to have reputation loss due to order picking failure.

Hong Kong/Macau and eastern China are more likely to have monetary loss arising from product nature failure than southern China. Hong Kong/Macau and eastern China companies are also more likely to have a loss from backorder loss from product nature than southern China. This situation also leads to an increase in cost. When there is product nature-related order fulfillment failure, Hong Kong/Macau companies are also more likely to have reputation loss, while those companies in eastern China also suffer more than those in southern China from the customer loss.

Companies that are privately owned are more likely to suffer from monetary loss, further backorder and increase in cost from product nature-related order fulfillment failure than the co-operated companies.

Local private companies are more likely to suffer from a lawsuit than the co-operated and state-owned companies due to order fulfillment failure from sustainability. Local private companies also suffer more than co-operated companies in term of monetary loss, reputation loss and loss of customers. Moreover, they are more likely to suffer from further backorder and an increase in cost due to sustainability order fulfillment failure than the state-owned companies.

Companies that are local private are more likely to suffer from lawsuit and monetary loss due to information sharing-caused order fulfillment failure, as stated. Local private companies also suffer from lawsuit due to the supply chain-caused order fulfillment failure more than the co-operated companies. Results show that sustainability and information sharing likelihood is significantly different among location groups. Particularly, the eastern part of China has a higher chance than Hong Kong/Macau and southern China of having the sustainability-related order fulfillment failure. Also, eastern China has a higher possibility of having the order fulfillment related to information sharing problem.

This research reports that logistics and sustainability likelihood are significantly different among management groups. It was found that full international companies are more likely to encounter order fulfillment failure arising from product nature (even it does not have overall significance) and sustainability. For the product nature, differences exist between the full and semi-international companies, while the differences in probability for sustainability-related order fulfillment exist between both the full international and semi-international/local companies.

Overall, there are significant differences in response time, product customization and target setting among various location groups. Environment, relationship with the business partners and the norms in the industry likelihood are different among different management styles.

Specifically, the routing is more likely to cause the order fulfillment failure for the state-owned company than the co-operated companies. Central/western China is more likely to have transport time-related order fulfillment failure than southern China. International companies are more likely to have order fulfillment failur caused by order tracking and transport time delay. Hong Kong/Macau, as well as the eastern side of China, are more likely to have product customization-related order fulfillment failure than the southern part of China. Full international companies are more likely to have order fulfillment failure related to environmental factors and their relationships with supply chain partners than the local companies. Semi-international companies are more likely to have norms in the industry-related order fulfillment failure than local companies.

Companies in Hong Kong/Macau are more likely to have order fulfillment failure caused by target setting than southern China and central/western China. Southern China also has a lower chance of having such failure than northern China. Northern China also has a far higher chance of having this failure than the central/western part of China. Domestic customer-based companies are more likely to suffer order fulfillment caused by make-to-order than the companies with an international customer base.

*4.4. Order fulfillment Paths in a Retail Supply Chain*

In this article, we focus on the retailer-side logistics of the order fulfillment process. Retailers have different fulfillment options that can be employed to fill customer demand (see Appendix A, Figure A8). These options include the use of existing DCs (**Note** Vendors (V), distribution centers (DC), direct-to-customer fulfillment centers (DTC), retail stores (R).) to fill online orders, known as integrated fulfillment [55]. In this case, retailers develop unified operational processes that combine warehousing activities for the store and online channels. The other option available to retailers is to use dedicated direct-to-customer order fulfillment centers. This dedicated fulfillment method requires a significant capital investment, process redesign, and coordinated product flows [56]. A critical threshold of online sales is needed to justify the high operating costs and inventory risks due to uncertain seasonal variations of demand in the online channel. Another approach used by retailers is to leverage inventory in local stores to fill online orders, known as store fulfillment [43]. If inventory investments and operational costs of supporting multiple channels are deemed prohibitive, retailers can also use their vendors to support online sales. In vendor fulfillment, retailers assign orders from online customers to vendors for fulfillment. The vendor fulfillment option allows retailers to sell additional products online without stocking inventory in the store and without the operational burden of filling online orders.

*4.5. Retail Distribution System*

This section presents a framework of the order fulfillment process in a retail distribution network (see Appendix A, Figure A9). In this framework, facilities are represented as nodes and arcs to indicate the flow of products and orders. Collectively, all nodes and arcs in the network are represented by set $\Omega$ (indexed by $\omega$) and set $E$ (indexed by $e$), respectively. The retailer operates several distribution facilities ($j \in \mathbb{D}$) and retail stores ($r \in \mathbb{R}$) in different markets ($m \in \mathbb{M}$). The assortment of products sold by the retailer is represented by ($k \in \mathbb{K}$) that have demand $\pi_t^{km}$ in market $m$ during a selling season ($t \in \mathbb{T}$). The proportion of demand in the online channel is represented by $p_t^{km}$. The retailer purchases products from vendors ($j \in \mathbb{V}$) for a unit cost-of-goods $\gamma_\omega^k$ for sale through $\omega$ that does not change through selling seasons. Note that $\gamma_v^k \geq \gamma_\omega^k : \omega \in \Omega \backslash \mathbb{V}$ because vendors do not offer a volume discount for smaller quantity orders shipped directly to customers [57].

Products flow in the retailer's distribution network over arcs represented by sets $E_I$ and $E_O$ for the inbound and outbound shipments, respectively. For each node $j$, sets $H_j$ and $T_j$ represent the arcs that start and end at node $j$, respectively. The quantity of product $k$ shipped over arc $e$ during time $t$ is given by the decision variables $X_{et}^k$. The unit inbound shipping cost for product $k$ is represented by the cost parameter $C_e^k : e \in E_I$ and unit outbound shipping cost is represented by $C_e^k : e \in E_O$. Each unit of product $k$ requires a storage space of $\alpha k$, whereas the total available storage space at the location $\omega \in \Omega$ is given by $k_\omega$. The end-of-season inventory $I_{\omega t}^k$ at location $\omega$ incurs inventory holding costs at the rate of $h_\omega^k$ per unit.

The framework incorporates four options to fill online demand (see Appendix A, Figure A9). These fulfillment options are based on the location of the inventory used to fill online orders. One of the options is to fill online orders from DC inventory (the same DCs that replenish retail stores), represented by set $\mathbb{D}$, that is, integrated fulfillment. The other option is for the retailer to operate separate direct-to-customer (DTC) fulfillment centers, represented by set $\mathbb{F}$ which serve online demand exclusively, that is, dedicated fulfillment. However, in this case, there are additional costs related to operating these facilities. We use a parameter $f_j \in F$ to represent the additional financial burden of operating separate fulfillment centers specifically for online orders. In the former case (integrated fulfillment), online orders are filled using existing DCs resulting in no additional fixed costs. In that sense, we use $f_j$ to signify the additional financial burden of operating separate fulfillment facilities over existing DCs. Another fulfillment option used by retailers is to fill online orders directly from vendors, represented by set $\mathbb{V}$ (i.e., vendor fulfillment). This option can be used by the retailer to sell an assortment of products in the online channel that it does not hold in stock. The fourth option is to

fulfill online orders from retail stores (i.e., store fulfillment), where orders are shipped directly from retail stores to online customers.

The retailer fills online demand from inventory held at any of the abovementioned stocking locations. The fulfillment and order delivery costs vary depending on the fulfillment node used. Based on these considerations, the best order fulfillment node can be identified. These choices are represented by decision variables $S_{\omega t}^{km}$ to identify the number of orders of product $k$ in market $m$ using inventory located at a node $\omega \in \Omega$, during selling season $t$.

The framework discussed above provides a useful mechanism to evaluate trade-offs in the order fulfillment process within the retail distribution. For instance, the fulfillment of online orders from retail stores may have the lowest last-mile delivery cost because of the stores' proximity to local customers. However, if orders are filled from retail stores, order picking and packing costs would be quite different compared to filling orders from DCs. Unlike a DC, the display shelves and inventory placement in a retail store are set according to marketing and sales considerations. These considerations may limit retailers' ability to efficiently pick items for order fulfillment, resulting in higher costs. Furthermore, the high real estate cost of urban and suburban retail outlets and significant inventory carrying costs to hold stock to fill online orders raise concerns.

A retailer may decide that concerns about high order fulfillment and inventory holding costs at retail stores warrant the use of existing DCs as order fulfillment nodes for the online channel. However, note that retailers' DCs are organized to efficiently stock or flow-through pallet-size or full-caseloads—a configuration not efficient for filling online orders. This dichotomy raises concerns about the suitability of using store-facing DCs for filling online orders.

The abovementioned dilemma can be resolved by using dedicated, direct-to-customer (DTC) fulfillment centers, which are configured exclusively for filling online orders. However, this option results in higher operating costs, as well as additional system-level inventory under the portfolio effect theory [58]. Hence, to achieve economies of scale, retailers favor using a few centrally located, large DTC fulfillment centers. The framework captures these trade-offs through inventory, transportation, warehousing operations, and delivery processes. The integrated decision as to which order fulfillment node is most suitable for a given scenario and how to incorporate inventory to support order fulfillment choices is included in the modeling framework. We used this framework and the mathematical representation of the retail distribution system described above to evaluate different order fulfillment options for retailers.

As per current practice, store-based retailers evaluate different fulfillment options to serve their customers based on the least cost-to-serve [59]. This criterion covers relevant supply chain activities, such as shipping products within the network of distribution facilities and stores, the order fulfillment process, inventory placement, and order delivery arrangements with third-party carriers. The following mathematical representation of the retail fulfillment network is developed in light of the framework discussed above for use with empirical data to identify the least cost-to-serve fulfillment options.

How We can Formulate this Model?

$\pi_t^{km} = $ *Demand of product* $k$ *in market* $m$ *in period* $t$,
$p_t^{km} = $ *Production of product* $k$ *demand online in market* $m$ *in period* $t$,
$\gamma_\omega^k = $ *Unit cost$-$of$-$goods for product* $k$ *sold through* $\omega$,
$C_e^k = $ *Unit cost$-$of$-$shipping* $k$ *through arc* $e \in E_I \cup E_O$,
$d_\omega^k = $ *Unit cost$-$of$-$filling an order of product* $k$ *from* $\omega$,
$\delta_\omega^{km} = $ *Unit cost of delivering an order of product* $k$ *from* $\omega$ *to market* $m$,
$k_\omega = $ *Storage allocation for inventory at location* $\omega$,
$h_\omega^k = $ *Unit inventory holding cost for* $k$ *at* $\omega$ *per period*,
$\alpha^k = $ *Unit storage requirements of* $k$,
$M = $ *Large number, and*
$f_i = $ *Operating cost per unit storage capacity at DTC* $j \in$ F.

*System variables* :

$X_{et}^k \geq 0$, *Units of **k** shipped through $e \in E_I \cup E_O$ in **t**,*

$Y_j = 1$, *if a DTC DC **j** is used for online channel,* 0 *otherwise, and*

$I_{\omega t}^k \geq 0$, *Inventory of product **k** held at $\omega$ at the end of **t**.*

*Decision variables* :

$S_{\omega t}^{km} \geq 0$, *Orders filled from $\omega$ for product **k** in market **m** during period **t**.*

$$(Q)\min \sum_{\omega \in \Omega,\, t \in T,\, k \in K,\, m \in M} S_{\omega t}^{km}\left(\gamma_\omega^k + d_\omega^k + \delta_\omega^{km}\right)$$
$$+ \sum_{j \in F} f_j(k_j)Y_j + \sum_{e \in E_o,\, t \in T,\, k \in K} C_e^k X_{et}^k + \sum_{\omega \in \Omega,\, t \in T,\, k \in K} h_\omega^k I_{\omega t}^k \tag{1}$$

$$Subject\ to: \quad \sum_{\omega \in \Omega} S_{\omega t}^{km} \leq p_t^{km}\left(\pi_t^{km}\right); \qquad \forall k \in \daleth,\ m \in \mathbb{M},\ t \in \mathbb{T}, \tag{2}$$

$$\sum_{e \in H_j} X_{et}^k = \sum_{m \in \mathbb{M}} S_{jt}^{km} + \sum_{e \in T_j} X_{et}^k + I_{jt}^k - I_{j(t-1)}^k; \qquad \forall j \in \mathbb{D},\ k \in \mathbb{K},\ t \in \mathbb{T}, \tag{3}$$

$$\sum_{e \in H_j} X_{et}^k = \sum_{m \in \mathbb{M}} S_{jt}^{km} + I_{jt}^k - I_{j(t-1)}^k; \qquad \forall j \in \mathbb{R} \cup \mathbb{F},\ k \in \mathbb{K},\ t \in \mathbb{T}, \tag{4}$$

$$\sum_{k \in \mathbb{K}} \alpha^k I_{\omega t}^k \leq k_\omega; \qquad \forall \omega \in \Omega \backslash \mathbb{V},\ t \in \mathbb{T}, \tag{5}$$

$$\sum_{k \in \mathbb{K},\, t \in \mathbb{T}} X_{jt}^k \leq MY_j; \qquad \forall j \in \mathbb{F}, \tag{6}$$

The objective of the model is to find the least cost fulfillment option. The first term of the objective Function (1) computes the total cost of order fulfillment, order delivery, and cost of goods for different fulfillment choices represented by decision variables $S_{\omega t}^{km}$. The second term is related to the retailer's decision to use dedicated DTC fulfillment center(s), which would add operating costs for such facilities. The third term computes the total outbound transportation costs from DCs to retail stores. The last term computes the total inventory holding cost of the system. The order fulfillment choices are made while considering several requirements related to supply, inventory and product sales. In Constraint (2), the model ensures online demand is filled from available inventory. The Constraints in (3) and (4) describe inbound/outbound product flows. The availability of products at DC $j$ is covered by supply from the vendors $\left(X_{et}^k\right)$ and the carryover inventory from the previous period $\left(I_{j,\,t-1}^k\right)$. The inbound supply variables $\left(X_{et}^k\right)$ are defined over a subset of edges $e \in T_j \subseteq \mathbb{E}_I$ associated with a fulfillment point $j$, which consists of all edges which terminate at $j$. The outbound flows in (3) include replenishment shipments from DCs to retail stores and orders sold/shipped to customers. Note that outbound product flows $X_{et}^k$ are defined over a subset $e \in H_j \subseteq \mathbb{E}_O$ associated with node $j$ that comprises all edges originating at $j$. Similar to DCs, product flow balance requirements are also described for retail stores and DTC fulfillment facilities through Constraint (4). Note that in (4), all outbound product flows are related to orders delivered to different markets. Constraints in (5) restrict inventory due to limited storage space. The constraints in (6) ensure that only operational DTC fulfillment centers are considered for the dedicated fulfillment option.

The model discussed above incorporates different fulfillment paths to serve online customers and fill orders using a mix of fulfillment options based on the relevant logistics costs and order delivery requirements. The model incorporates the underlying differentiation of fulfillment options by considering: (i) order fulfillment costs that differ across facilities due to labor wages, facility layouts, and variations in pick-pack-ship operations; (ii) online orders that have different (last-mile) delivery requirements; (iii) inventory costs that vary across echelons (i.e., store vs. DCs); and (iv) that there are delivery considerations such as pick-from-store and free shipping. Our model captures these issues

through model parameters, grounded in empirical data and the model/analysis setting developed through discussions with the retail supply chain executives consulted for this research.

The modeling framework and the resulting mathematical representation of the retail distribution system discussed above were used with data collected from a large store-based retail firm in the United States. These data (described in the next section) allowed us to create a base-case setting representing the legacy distribution network used by the retailer. Using these data within a factorial design research study, we evaluated order fulfillment options through the analysis of different scenarios and the evaluation of trade-offs among order fulfillment, order delivery, inventory, and transportation.

*4.6. Concerns and Performance Focus Lead to the Order Fulfillment Failure*

In this section, the respondents would be asked to rank their concerns, state their performance in particular performance to see whether the particular focus (both willingness and actual performance) will lead to higher or lower chance/loss in a particular order fulfillment failure.

For the willingness to focus on a particular performance indicator, the ranking is used. These concerns (ranking) will be put into the list of the independent variables, and the likeliness/loss will be put into the dependent variable.

From Tables A1–A3, it is found that the order picking, sustainability, and supply chain-related order fulfillment failures are somehow related to the focus and concern over various performance indexes. For the order picking order fulfillment failure, the higher the concern over the documentation accuracy, the less likely the failure will occur. For the sustainability-related failure, focus on certain performance indexes increases the likelihood of the failure. For instance, the more concern over the documentation accuracy, response toward the failure and the cost increase due to order fulfillment, the more likely the company will suffer the sustainability-related order fulfillment failure. On the other hand, the supply chain-related order fulfillment failure is likely to occur when the company puts more focus on the delay issue.

For delay control in order fulfillment, the larger the loss from the logistics order fulfillment failure, the better the delay control. On the other hand, the lower the loss from sustainability-related order fulfillment failure, the better the delay control performance. For the likelihood, the lower the chances of having the information sharing-related order fulfillment failure, the tighter the control. However, the higher the chances of having the supply chain order fulfillment failure, the better performance in the delay control (see Table A4). A similar situation applies to the documentation accuracy level. High loss from logistics-related failure and small loss from the sustainability order fulfillment failure mean the higher the documentation accuracy. The lower the chances of having the logistics failure, the better the documentation. Surprisingly, the higher possibility of information sharing failure will lead to good documentation (see Table A5). The only factor that affects the rate of the product in perfect condition is the order picking failure likeliness. For instance, the lower the chances of having such a failure, the higher the rate of goods in perfect condition (see Table A6).

The higher the loss from logistics-related failure and the smaller the loss from the sustainability order fulfillment failure, the better the response time (see Table A7). Results report that there are no order fulfillment-related issues that are related to the cost increase control and service level (see Tables A8 and A9).

## 5. Discussion

*5.1. Qualitative Research*

The major objective of qualitative research is to confirm the risk factors that were used in previous research studies across different industries. This is because there is no structured research on order fulfillment in the Disposable Products Industry. Companies participating in the qualitative research are of different scales and sizes to see whether there are any special risk factors specific to a sector or sections.

The risk factors that are listed by the respondents can all be put into the predefined categories. Particularly, the respondents said that people should pay more attention to the environment-related factors that are mostly caused by policies that are subject to change from time to time. It is interesting to know that the environmental concerns may provide an opportunity to the Disposable Products Industry, especially those companies that can provide new Disposable Products. That means the failure of one company to fulfill the order may be the chance for the other companies who are prepared for the changes in the long run. Order fulfillment may not be solely negative [60].

It is important for a company to pay attention to the quality of the materials supplied to the company as that would be one of the major sources of the internal order fulfillment failure. To correct such problem it is much more important to have a better relationship with the suppliers in order to have better monitoring of the material and thus the order fulfillment.

It is also important to know that the Disposable Products Industry should look into details of the order fulfillment problems by products involved instead of the position of the company along the chain. Even with the possible differences in the causes of order fulfillment failure, the delivery time concern is still the major and most important measurement in the order fulfillment. This is because time is still the most quantifiable measurements for the performance.

*5.2. Quantitative Research*

5.2.1. Loss in Order Fulfillment

Companies that are based in the eastern part of China and with the international customer base should pay more attention to the loss arising from the order fulfillment failure. This is because the companies who are working with customers around the world with international standards are more complex and thus suffer higher loss from the order fulfillment arising from sustainability. The suability loss problem is also served by the complex structure of the company itself (co-operated). Because of the complexity more focus should be put not only on management but the broad consequences of the actual loss to the lawsuit, increase in cost in general, further backorder and long-term reputation loss, which is hardly found in any other fulfillment failures.

Also, companies in various parts of China should pay attention to a different source of order fulfillment loss in the northern part of China. For instance, in the northern part of China, it is more important to focus on the order picking while the other parts of China (Hong Kong/Macau and eastern part) should focus more on the product complexity.

5.2.2. Likelihood of Order Fulfillment

For the likelihood, the company should pay more attention to the product nature; even though it may not be as harmful as sustainability in terms of actual loss, it is found that the nature of the company's product does lead to differences in the likelihood. Again, international companies with more complex management and customer needs would face different product requirements. However, it is found that product nature and sustainability problems usually happen together for the international firms, as both may arise from the compliance with standards and procedures from customers and suppliers from all over the world. This may be the reason why companies usually are more likely to face the order fulfillment products related to these aspects at the same time. The results also match with those provided in the qualitative research as respondents believe that the relationship with the suppliers which also put into the sustainability aspect is one of the key factors leading the material quality variations and thus the order fulfillment failure. Therefore, it is very important to have better information sharing and good relationships with the various parties along the chain.

At the micromanagement level, companies which are not co-operative should focus more on transportation and routing; this problem is also significant to those with relatively new developing road networks like those in the central and western parts of China.

### 5.2.3. Performance References

It is important to know that the concerns and focus on improving the company performance lead to better results in the order fulfillment. It is found that the concerns may provide good or bad results for the companies. For instance, the attention over documentation accuracy does pay off in terms of reducing the chances of having order picking-related order fulfillment failure. This is obvious in that it would directly reduce the chance of picking the wrong order and thus making wrong delivery to the customers. However, this focus together with similar short-term relief (cost reduction and response time), which successfully reduces the operational level order fulfillment problem, may decrease the resources that can lower the chances of having long-term sustainability-related order fulfillment failure.

The continuous emphasis on the sustainability factors is justifiable as the ability to control the loss arising from this can in the short term solve the delay in order fulfillment in the real term. Instead of having very tight documentation accuracy control, it is better to keep a good relationship with the customers and the suppliers, as this can, in the long run, lead to more accurate documentation. This may be due to the co-operation in improving the documentation transfer rather than doing it separately by the company.

### 5.2.4. Order Fulfillment: Fractional Multinomial Logit Model

The results were analyzed to evaluate the effect of order fulfillment, order delivery, warehousing, shipping, and inventory-related factors on dependent variables (proportion of orders filled by each fulfillment option). The nature of our data required fitting a model that can identify the proportion of orders filled by each of the four fulfillment options. Thus, dependent variables in our study are not continuous, rather they are proportions with values between 0 and 1 that collectively sum to 1 for each data observation. For this type of data, traditional estimation methods (linear regression, general linear method, etc.) were not appropriate [61].

To analyze data with a fractional response variable, the fractional multinomial logit model (FMlogit) technique was used [62–64]. This technique combines multivariate fractional logit and multinomial logit such that the model output represents the expected values of the proportions for different response variables, all of which sum to 1. In our case, proportions represent the use of different fulfillment options to fill online orders. The FMlogit model measures the changes in multiple response variables simultaneously as a measure of the effects of independent variables. Consider a random sample of $i = 1, ..., N$ observations of orders, each with $M$ outcomes of fulfillment choices. In our study, $M = 4$, which corresponds to each of the order fulfillment types. Given $s_{ik}$ represents the kth outcome for observation i, and $x_i$, $i = 1, ..., N$, is a vector of exogenous covariates. The FMlogit model assumes that $M$ conditional means have a multinomial logit functional form in linear indices as:

$$E[S_k|x] = G_k(x; \beta) = \frac{\exp(x\beta_k)}{\sum_{m=1}^{M} \exp(x\beta_m)}, \quad (7)$$

We can define a multinomial logit quasi-likelihood function $L(\beta)$ that uses the observed shares $S_{ik} \in [0, 1]$ in place of a binary indicator that would otherwise be used in multinomial functions:

$$L(\beta) = \prod_{i=1}^{N} \prod_{m=1}^{M} G_m(x_i; \beta)^{S_{im}}, \quad (8)$$

The parameters of the FMlogit model are obtained using a quasi-maximum likelihood estimator for the fractional response model without an ad hoc transformation of boundary values [65]. Because this method accounts for all outcome options that add up to 1, there is no issue of violating the Independence of Irrelevant Alternatives (IIA) assumption. The statistical significance of the overall FMlogit model is tested through the chi-square test statistic. This statistic evaluates the null hypothesis that none of the independent variables affect the response variable. The null hypothesis is rejected for

a *p-value* less than the level of significance. Similarly, the statistical significance of the independent variables is tested through the *z-statistic* and its *p*-value, which corresponds to the null-hypothesis that the respective terms in the model are equal to zero. A *p-value* less than the 0.05 significance level provides statistical evidence to reject the null hypothesis.

Our data set consisted of 6561 combinations of independent variable values. The corresponding choice of the fulfillment option is represented by the dependent variables: DTC[p], DC[p], R[p], and V[p]. Table A10 summarizes some basic descriptive statistics about the response variables. We used the STATA code developed by Buis (2008) for estimating the parameters of the FMLogit model. The output of the model for beta coefficients, z-statistic and its p-value was recorded for all levels of the response variables. The test statistics reported in Table A11 (see the last row) indicate that the factional multinomial model converged on a log pseudolikelihood of −3064.97 with a Wald chi-squared of 23,869. A high fit between data and predicted values from the model was recorded (*Prob* > $\chi 2$ = 0.000). Average marginal effects that were statistically different from zero at 1% and 5% levels are indicated with two and one asterisks, respectively.

Each coefficient in Table A11 represents the marginal change in fulfillment options as a result of a unit change in the independent variables. For example, the marginal effect for $d_\mathbb{F}$ under DTC[p] is −0.0915, which suggests that a unit increase in DTC fulfillment cost is associated with a decrease of 9.15% in the use of the DTC fulfillment option. Because all fulfillment options must sum to 1, this shift would mean that some orders will be filled by other fulfillment options. The results show that a big share of these orders will be filled using DC fulfillment (coefficient = 0.0852). Similarly, a unit increase in the vendor fulfillment fee $d_\mathbb{V}$ would result in a reduction of 6.29% in the use of the vendor fulfillment option V[p]. These orders are shifted to DTC for fulfillment (coefficient = +0.0697).

The results show that orders that are filled through DCs and DTCs would shift between these fulfillment options under a marginal effect due to changes in relevant (fulfillment and delivery) costs. This effect is seen for $d_\mathbb{F}$, $f_\mathbb{F}$ *and* $\delta_\mathbb{F}$ variables. For example, a unit change in $f_\mathbb{F}$ would shift an equal proportion of orders from DTC fulfillment (−7.31%) to DC fulfillment (+6.88%). It is also interesting to note that while both fulfillment costs and order delivery costs show a significant effect on the choice of order fulfillment option, the scale of effect for fulfillment costs is much higher than that for order delivery costs. For example, a unit increase in delivery costs $\delta_\mathbb{F}$ would reduce DTC fulfillment by 5.69% whereas the unit change in fulfillment costs would reduce DTC use by 9.15%. We explore these dynamics in more depth later.

The marginal effects for the order delivery costs are more prominent on the vendor fulfillment option than the store fulfillment options. A unit increase in delivery costs reduces the use of store fulfillment by 1.23% and vendor fulfillment by 5.02%. The small scale of the effect on store fulfillment can be explained by the small use of this fulfillment option for most combinations of factor levels in the factorial design. We investigate this aspect in more detail later.

The discussion above is based on a full-model analysis using the fractional multinomial logit model approach. This analysis identified the effects of different factors on the response variables: DTC[p], DC[p], R[p], and V[p]. In the next step of the analysis, we focused specifically on two key elements: order fulfillment and order delivery, in terms of how changes in these factors would affect the retailer's use of respective fulfillment options. The objective of this detailed analysis (discussed later) was to seek managerial insights regarding cost thresholds to understand shifts in the use of order fulfillment options. The interpretation of these shifts is meaningful via comparison of different scenarios when the retail firm operates under its current setting, as represented by the base case. This understanding will guide retailers to set operational targets to achieve certain cost thresholds to promote or sustain the use of different fulfillment options.

The results (see Table A12) show that the use of the DTC fulfillment option depends greatly on warehousing and order fulfillment costs. DTC fulfillment was the primary option (100% orders filled) for warehousing cost levels $f_\mathbb{F} \leq \$10.00$ and fulfillment cost level $d_\mathbb{F} = \$5.25$. However, when DTCs operate at a higher cost level $f_\mathbb{F} \geq \$15.00$, the differential in fulfillment costs between existing DCs and

new DTCs would favor the former. For example, DTCs filled only 25% of total orders even when DTC unit order fulfillment cost ($f_{\mathbb{F}} = \$5.25$) was 25% less than the DC fulfillment option $\left(i.e., \frac{d_{\mathbb{F}}}{d_{\mathbb{D}}} = 0.75\right)$. None of the orders was processed by DTCs when fulfillment cost differential between DTC and DC options was reduced to 5%, that is, $\frac{d_{\mathbb{F}}}{d_{\mathbb{D}}} = 0.95$ (see the last row in Table A12).

Note that a lower $d_{\mathbb{F}}$ level in Table A12 is related to a setting in which the operational efficiency of DTC fulfillment centers is correspondingly higher than a DC. For example, $d_{\mathbb{F}} = \$5.25\left(\frac{d_{\mathbb{F}}}{d_{\mathbb{D}}} = 0.75\right)$ level represents the case where DTC order fulfillment is 25% better than the base-case DC fulfillment $d_{\mathbb{D}} = \$7.00$ level. This gain is achieved in a DTC fulfillment center by focusing resources and business processes on the particular order profile of online orders, which is quite different from store replenishment orders handled by DCs. The results in Table A12 show that DTC remained the best fulfillment option until the marginal difference between DTC and DC order fulfillment costs reached 85% $\left(\frac{d_{\mathbb{F}}}{d_{\mathbb{D}}} = 0.85\right)$. This setting ($d_{\mathbb{F}} = \$5.95$), 35% of online demand is filled from store-facing DCs, whereas all online orders were filled by store-facing DCs for $d_{\mathbb{F}} = \$6.65$. To justify the use of DTC fulfillment, retailers have to consider a balance between order fulfillment efficiency and facility operating costs. For example, to compensate for 50% higher DTC operating costs ($f_{\mathbb{F}} = \$10.00$ level vs. $f_{\mathbb{F}} = \$15.00$ level), full utilization of DTC fulfillment (100% orders) would require a DC order fulfillment cost deferential of more than $+30\%\left(i.e., d_{\mathbb{F}} \leq \$5.00; \frac{d_{\mathbb{F}}}{d_{\mathbb{D}}} \leq 0.70\right)$.

## 6. Conclusions and Further Studies

This research presented a detailed framework of the order fulfillment process in the retail supply chain. This framework considered multiple fulfillment options available to store-based retailers that include existing network of stores and distribution (DC) facilities, new direct-to-customer (DTC) fulfillment centers, and collaborating with vendors to fill online orders directly from vendor facilities. In an empirical study, these fulfillment options were evaluated to provide managerial insights regarding each fulfillment option and identify operational and cost thresholds where a particular fulfillment option would be preferred.

Using data from a large U.S. retailer and a factorial design approach, this study found that large distribution facilities (DCs and DTCs) provide economies of scale more suitable for order fulfillment compared to other options. However, retailers would need to carefully consider the corresponding increase in operating costs in their DC/DTC networks. The study identified an efficiency threshold, reflected by the 10% differential in order fulfillment costs, for the DTC option to justify operating dedicated facilities to fill online orders. Conversely, retail firms that can operate DTC facilities with low operating costs would not need to achieve a prohibitively high order fulfillment efficiency goal to justify the use of DTC facilities [66–68]. The study also indicated that the usefulness of an order fulfillment option depends greatly on the corresponding order delivery process. For example, in our study, a 15% discount on order delivery charges allowed the retailer to use existing DCs and avoid the need to operate dedicated (DTC) fulfillment facilities.

This study found that stores are costly order fulfillment nodes for retailers compared to the distribution facilities. Retailers would need to improve store fulfillment (through improved store processes and better training of store associates) to bring the combined fulfillment and order delivery costs under USD 10.00 per order. Above this threshold, store fulfillment is not a very competitive option compared to DCs and DTCs. The results also highlighted that it is important for retailers to allow sufficient store inventory for online orders to maximize the use of store fulfillment, especially for the pick-from-store delivery option. The study also found that a lower order processing/fulfillment fee, charged by vendors, would be needed (reduced to around USD 3.00 per order) to make vendor fulfillment a cost-competitive fulfillment option for retailers. Another approach that would favor vendor fulfillment is to negotiate discounted delivery rates with package carriers so that retailers' total cost per order is below the USD 6.40 threshold.

Overall, this study presented a detailed analysis of different settings for the order fulfillment and order delivery process and demonstrated how changes in the underlying cost structure can impact the usefulness of one fulfillment option over another. The study concluded that the scale economies achieved in DC and DTC fulfillment facilities outperform the store and vendor fulfillment options. However, with some changes in the order fulfillment and delivery processes, retailers can leverage their stores more effectively for order fulfillment.

From this study, it was found that the risk factors encountered by the Disposable Products Industry leading to order fulfillment failure were much similar than those in the other industries. This was confirmed by the qualitative and quantitative part of this study in which no answer failed to be grouped to the existing categories provided by previous research studies from other industries.

To summarize the results of the study, sustainability is no doubt the most important factor that needs attention in the Disposable Products Industry. This is because companies can incur many types of tangible and intangible losses through the order fulfillment failure. This is particularly important for those who are focusing on the international market.

Also, sustainability, which can be referred to as the ability to maintain long-term relationships with the customers and other parties, may be intensified as it is more likely to affect the international firms as mentioned above, which may follow different standards and product requirements. Transportation has a high ability to occur in the local context.

More importantly, companies in the Disposable Products Industry should note that over concerns about the particular order fulfillment criteria may not help the situation; to be exact, the focus on short-term performance accuracy may increase the long-term order fulfillment failure. It is better to focus on how to reduce the order fulfillment failures in the long run to improve the situation as a whole, rather than giving up a chance for bettering the company performance in a long-lasting manner. To conclude, it is better to focus on the long-term order fulfillment problem rather than focus on a particular performance indicator.

The current study is only the first step in the order fulfillment analysis for the highly diversified Disposable Products Industry. The following ideas may be good directions for further investigations in the foreseeable future.

- It is interesting to break down Disposable Products into various groups as the respondents in the qualitative suggested that the product nature may affect the order fulfillment failure.
- The current research contrasts the differences in order fulfillment likelihood, loss and performance for various parts of China, and it may be a good idea to extend the analysis to different parts of the world, especially since internationalization will impose a difference on the current study.
- The current study suggests that it is more important to focus on the long-term order fulfillment factors, and it may be reasonable to provide a longitudinal study to contrast the long-term order fulfillment performances for companies with different levels of attention towards this long-term factor.

**Author Contributions:** Conceptualization; methodology; software; validation; formal analysis; investigation; resources; data curation; writing—original draft preparation; writing—review and editing; visualization, M.H.; supervision, X.Z.; project administration; funding acquisition, K.K.L. All authors have read and agreed to the published version of the manuscript.

**Funding:** This research received no external funding.

**Conflicts of Interest:** On behalf of all authors, the corresponding author states that there is no conflict of interest.

**Appendix A**

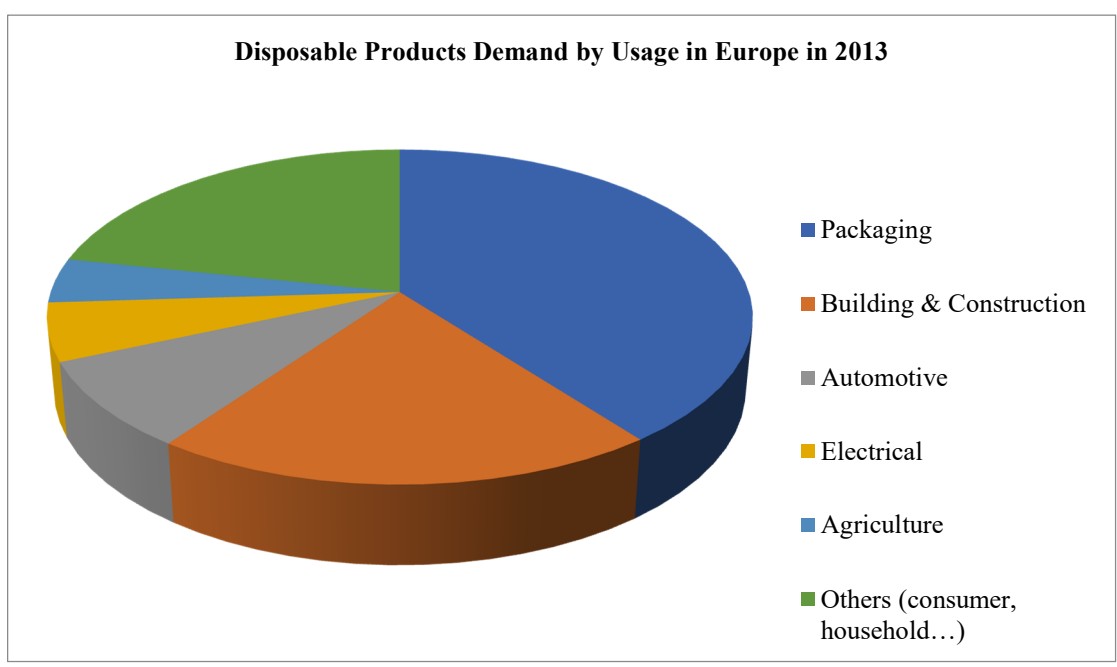

**Figure A1.** Disposable Products Demand by Usage in Europe in 2013 [55].

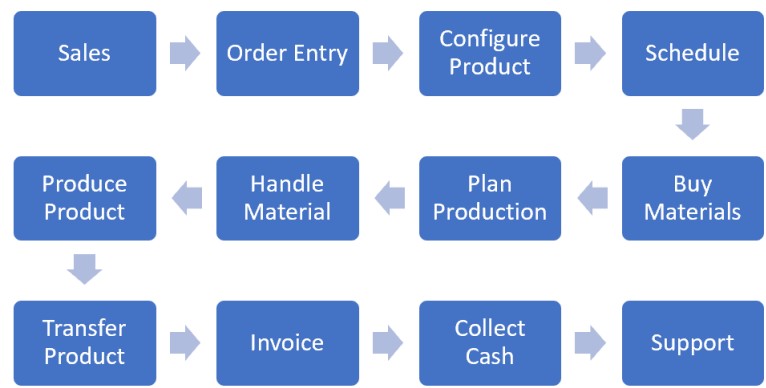

**Figure A2.** Value stream defined order fulfillment process [9].

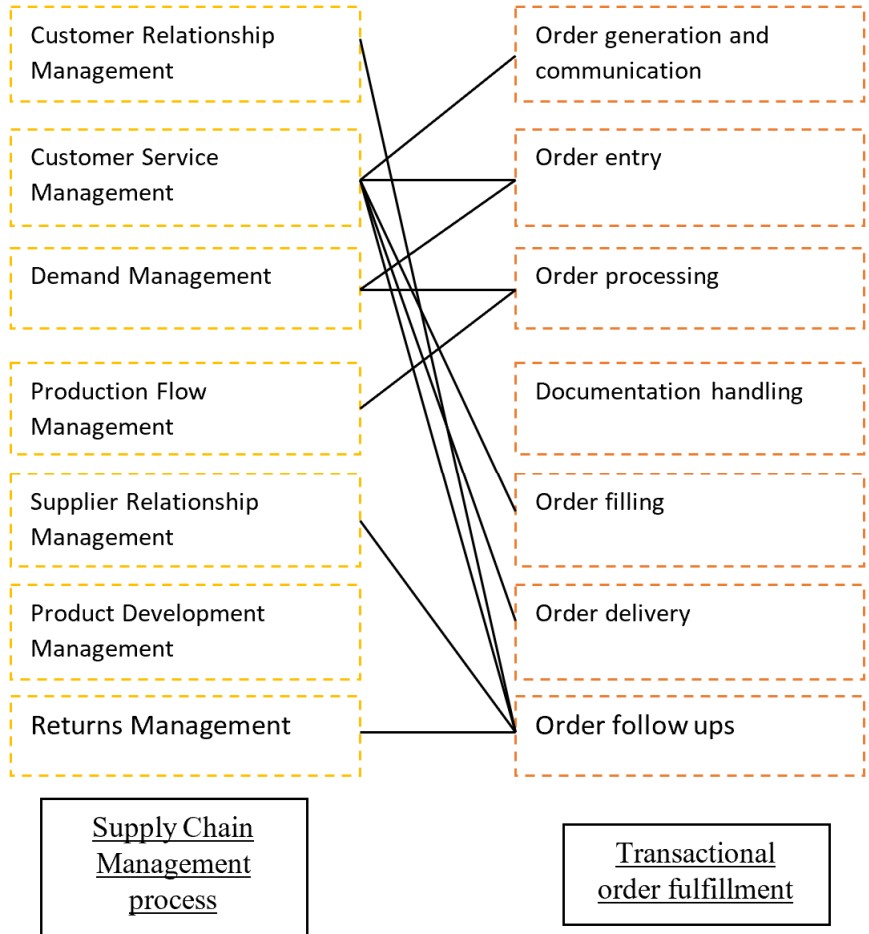

**Figure A3.** Transactional order fulfillment [13].

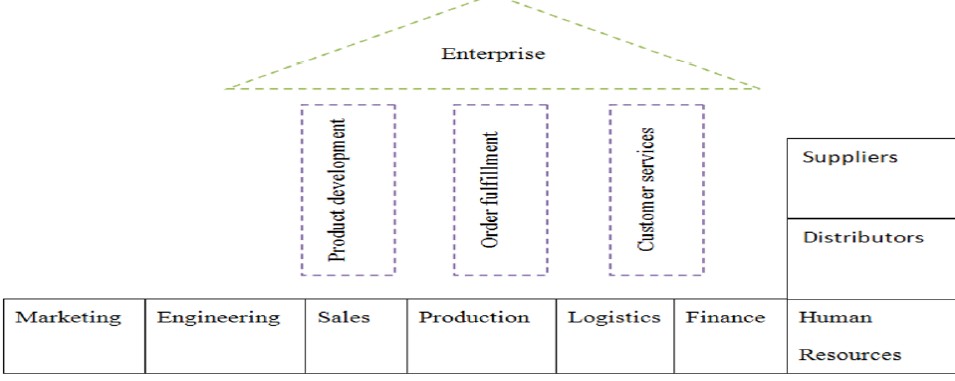

**Figure A4.** Relationship of the order fulfillment with the other process and entities [11].

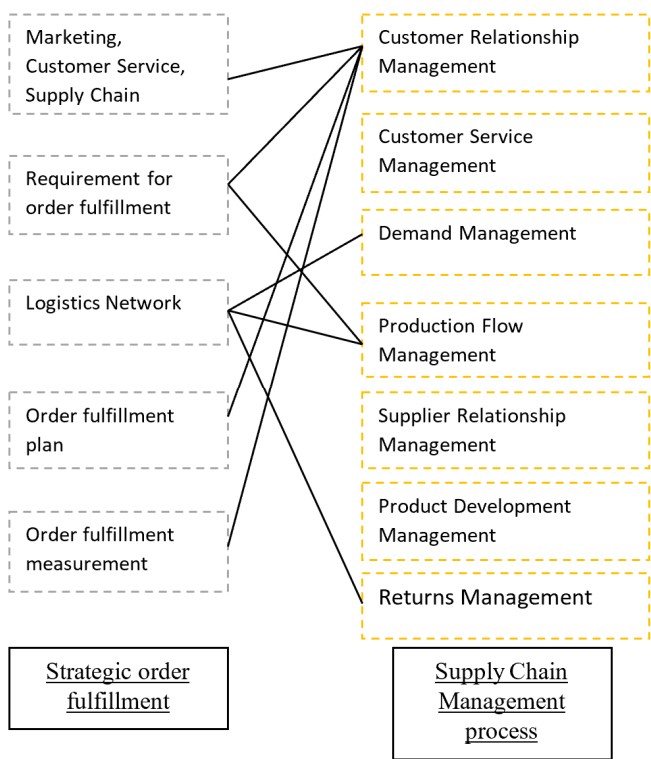

**Figure A5.** Strategic order fulfillment process and supply chain management process [13].

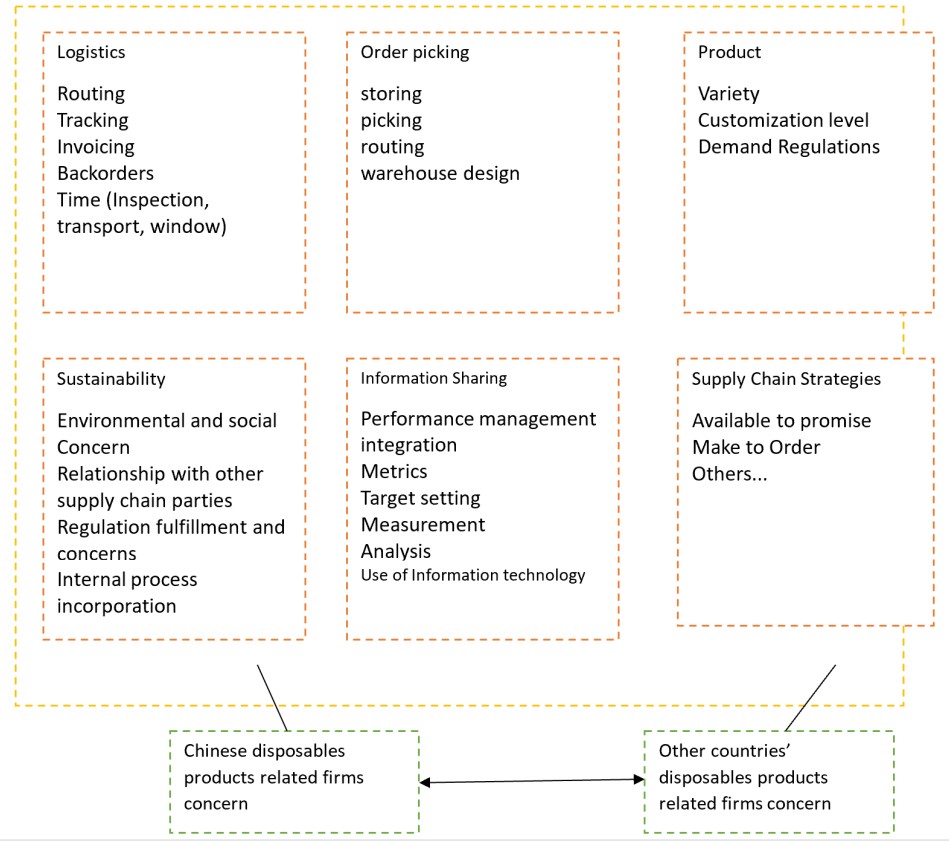

**Figure A6.** Concerns comparison among the Chinese and other countries' disposable products firms.

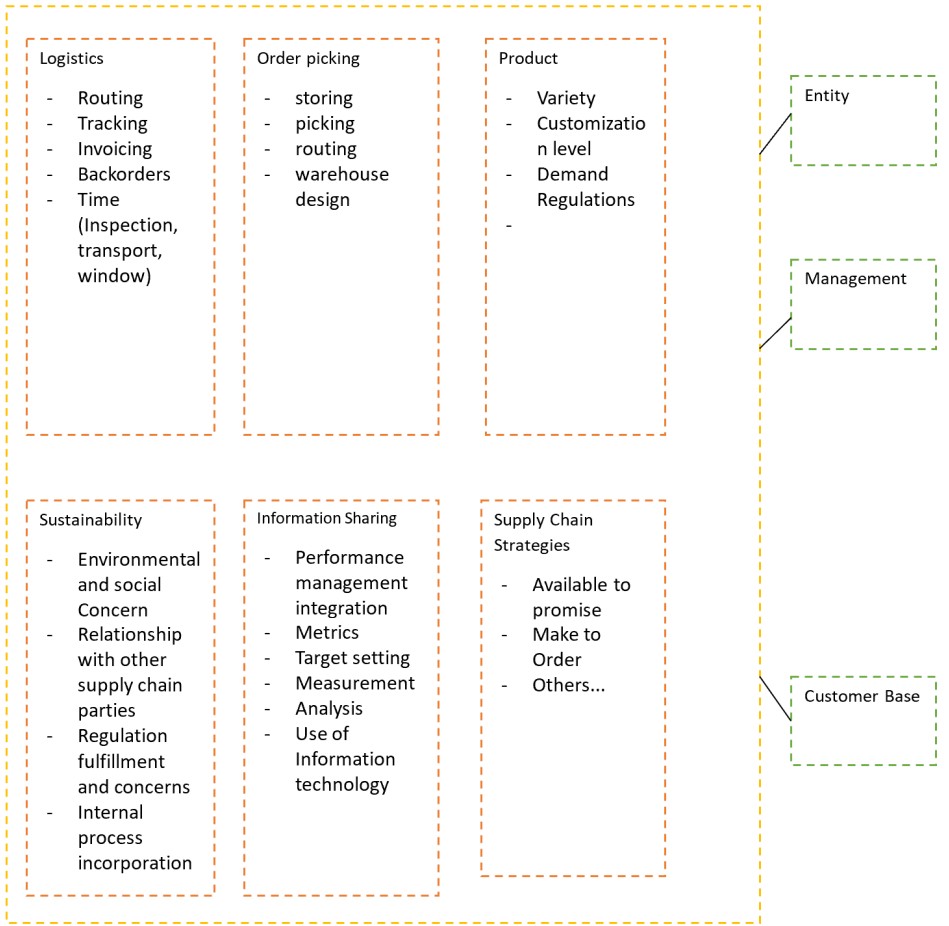

**Figure A7.** Concerns comparison among various layers and parties of the disposable products industries.

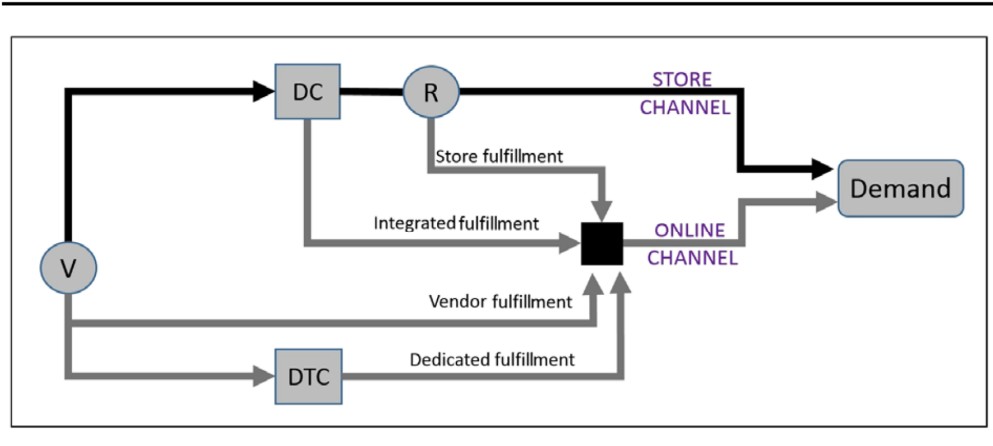

**Figure A8.** Order fulfillment paths in a retail supply chain.

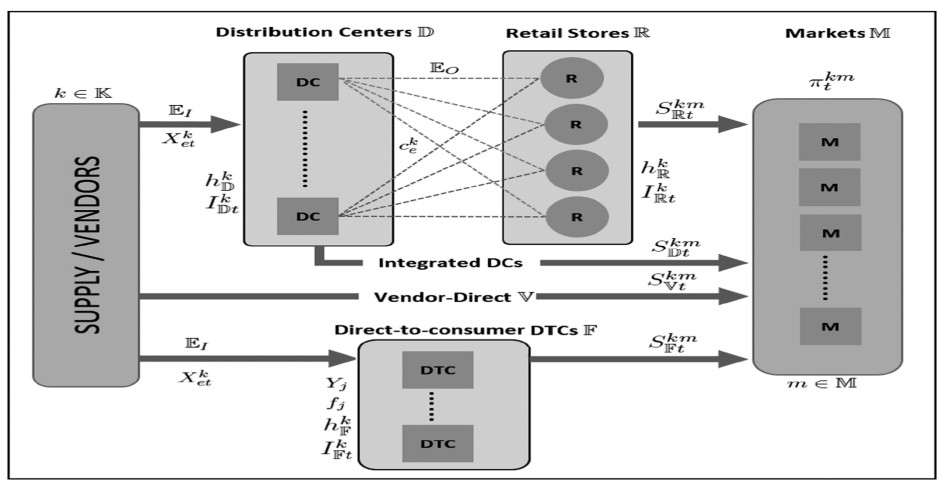

**Figure A9.** Order fulfillment framework.

## Appendix B

**Table A1.** Model significance for the regression on ranking and order picking order fulfillment failure likelihood.

| | | Coefficients[a] | | | | |
|---|---|---|---|---|---|---|
| **Model** | | **Unstandardized Coefficients** | | **Standardized Coefficients** | **t** | **Sig.** |
| | | **B** | **Std. Error** | **Beta** | | |
| 1 | (Constant) | 1.645 | 0.577 | | 2.850 | 0.005 |
| | Delay (Rank) | 0.013 | 0.067 | 0.019 | 0.201 | 0.841 |
| | Doc (Rank) | 0.130 | 0.062 | 0.203 | 2.118 | 0.036 |
| | Condition (Rank) | 0.003 | 0.070 | 0.004 | 0.039 | 0.969 |
| | Response (Rank) | −0.004 | 0.080 | −0.005 | −0.050 | 0.960 |
| | Cost (Rank) | 0.075 | 0.065 | 0.113 | 1.147 | 0.254 |
| | Service (Rank) | 0.014 | 0.055 | 0.025 | 0.251 | 0.803 |

Note: Coefficients[a] means Coefficient$^{0.05\%}$.

**Table A2.** Model significance for the regression on ranking and sustainability order fulfillment failure likelihood.

| | | Coefficients[a] | | | | |
|---|---|---|---|---|---|---|
| **Model** | | **Unstandardized Coefficients** | | **Standardized Coefficients** | **t** | **Sig.** |
| | | **B** | **Std. Error** | **Beta** | | |
| 1 | (Constant) | 4.939 | 0.593 | | 8.324 | 0.000 |
| | Delay (Rank) | −0.124 | 0.071 | −0.162 | −1.751 | 0.083 |
| | Doc (Rank) | 0.073 | 0.064 | 0.105 | 1.135 | 0.259 |
| | Condition (Rank) | −0.049 | 0.072 | −0.064 | −0.678 | 0.499 |
| | Response (Rank) | −0.238 | 0.082 | −0.260 | −2.889 | 0.005 |
| | Cost (Rank) | −0.140 | 0.068 | −0.196 | −2.076 | 0.040 |
| | Service (Rank) | −0.036 | 0.057 | −0.059 | −0.632 | 0.529 |

Note: Coefficients[a] means Coefficient$^{0.05\%}$.

**Table A3.** Model significance for the regression on ranking and supply chain order fulfillment failure likelihood.

| | | Coefficients[a] | | | | |
|---|---|---|---|---|---|---|
| | **Model** | **Unstandardized Coefficients** | | **Standardized Coefficients** | **t** | **Sig.** |
| | | **B** | **Std. Error** | **Beta** | | |
| 1 | (Constant) | 4.155 | 0.596 | | 6.974 | 0.000 |
| | Delay (Rank) | −0.131 | 0.068 | −0.189 | −1.942 | 0.055 |
| | Doc (Rank) | 0.058 | 0.063 | 0.089 | 0.913 | 0.363 |
| | Condition (Rank) | −0.045 | 0.072 | −0.062 | −0.619 | 0.537 |
| | Response (Rank) | −0.016 | 0.081 | −0.018 | −0.192 | 0.848 |
| | Cost (Rank) | −0.081 | 0.068 | −0.120 | −1.195 | 0.235 |
| | Service (Rank) | −0.005 | 0.056 | −0.009 | −0.091 | 0.928 |

Note: Coefficients[a] means Coefficient$^{0.05\%}$.

**Table A4.** Model significance for the regression on failure and delay performance.

| | | Coefficients[a] | | | | |
|---|---|---|---|---|---|---|
| | **Model** | **Unstandardized Coefficients** | | **Standardized Coefficients** | **t** | **Sig.** |
| | | **B** | **Std. Error** | **Beta** | | |
| 1 | (Constant) | 3.086 | 0.412 | | 7.481 | 0.000 |
| | Logistics | 0.167 | 0.098 | 0.204 | 1.697 | 0.093 |
| | Order picking | −0.132 | 0.108 | −0.147 | −1.219 | 0.226 |
| | Product nature | 0.130 | 0.103 | 0.147 | 1.265 | 0.209 |
| | Sustainability | −0.262 | 0.117 | −0.324 | −2.245 | 0.027 |
| | Information sharing | 0.218 | 0.133 | 0.254 | 1.636 | 0.105 |
| | Supply chain strategies | 0.057 | 0.130 | 0.069 | 0.440 | 0.661 |
| | Logistics (Likely) | −0.047 | 0.090 | −0.063 | −0.521 | 0.603 |
| | Order picking (likely) | −0.125 | 0.109 | −0.134 | −1.150 | 0.253 |
| | Product nature (likely) | 0.030 | 0.096 | 0.040 | 0.308 | 0.759 |
| | Sustainability (likely) | 0.149 | 0.103 | 0.182 | 1.442 | 0.153 |
| | Information sharing (likely) | −0.222 | 0.126 | −0.251 | −1.755 | 0.082 |
| | Supply chain (likely) | 0.189 | 0.113 | 0.231 | 1.670 | 0.098 |

Note: Coefficients[a] means Coefficient$^{0.05\%}$.

**Table A5.** Model significance for the regression on failure and documentation accuracy.

| | | Coefficients[a] | | | | |
|---|---|---|---|---|---|---|
| | **Model** | **Unstandardized Coefficients** | | **Standardized Coefficients** | **t** | **Sig.** |
| | | **B** | **Std. Error** | **Beta** | | |
| 1 | (Constant) | 3.525 | 0.413 | | 8.546 | 0.000 |
| | Logistics | 0.217 | 0.102 | 0.260 | 2.116 | 0.037 |
| | Order picking | −0.132 | 0.109 | −0.145 | −1.215 | 0.227 |
| | Product nature | 0.134 | 0.103 | 0.149 | 1.297 | 0.198 |
| | Sustainability | −0.105 | 0.116 | −0.127 | −0.905 | 0.368 |
| | Information sharing | −0.107 | 0.131 | −0.123 | −0.818 | 0.415 |
| | Supply chain strategies | 0.103 | 0.141 | 0.121 | 0.732 | 0.466 |
| | Logistics (Likely) | −0.171 | 0.094 | −0.225 | −1.812 | 0.073 |
| | Order picking (likely) | −0.120 | 0.111 | −0.128 | −1.078 | 0.284 |
| | Product nature (likely) | −0.105 | 0.105 | −0.135 | −1.003 | 0.318 |
| | Sustainability (likely) | 0.180 | 0.104 | 0.213 | 1.738 | 0.085 |
| | Information sharing (likely) | 0.136 | 0.125 | 0.150 | 1.093 | 0.277 |
| | Supply chain (likely) | 0.105 | 0.117 | 0.127 | 0.894 | 0.374 |

Note: Coefficients[a] means Coefficient$^{0.05\%}$.

**Table A6.** Model significance for the regression on failure and product in perfect condition.

| | Coefficients[a] | | | | |
|---|---|---|---|---|---|
| **Model** | **Unstandardized Coefficients** | | **Standardized Coefficients** | **t** | **Sig.** |
| | **B** | **Std. Error** | **Beta** | | |
| (Constant) | 3.314 | 0.440 | | 7.540 | 0.000 |
| Logistics | 0.121 | 0.106 | 0.141 | 1.139 | 0.257 |
| Order picking | −0.183 | 0.115 | −0.198 | −1.597 | 0.114 |
| Product nature | 0.161 | 0.109 | 0.174 | 1.473 | 0.144 |
| Sustainability | −0.089 | 0.125 | −0.106 | −0.710 | 0.480 |
| Information sharing | −0.036 | 0.143 | −0.040 | −0.255 | 0.800 |
| Supply chain strategies | 0.060 | 0.150 | 0.069 | 0.402 | 0.689 |
| Logistics (Likely) | 0.056 | 0.099 | 0.072 | 0.568 | 0.571 |
| Order picking (likely) | −0.219 | 0.119 | −0.227 | −1.833 | 0.070 |
| Product nature (likely) | −0.074 | 0.112 | −0.092 | −0.660 | 0.511 |
| Sustainability (likely) | 0.157 | 0.110 | 0.181 | 1.427 | 0.157 |
| Information sharing (likely) | 0.021 | 0.135 | 0.023 | 0.156 | 0.876 |
| Supply chain (likely) | 0.126 | 0.127 | 0.149 | 0.990 | 0.325 |

Note: Coefficients[a] means Coefficient$^{0.05\%}$.

**Table A7.** Model significance for the regression on failure and response time.

| | Coefficients[a] | | | | |
|---|---|---|---|---|---|
| **Model** | **Unstandardized Coefficients** | | **Standardized Coefficients** | **t** | **Sig.** |
| | **B** | **Std. Error** | **Beta** | | |
| (Constant) | 3.209 | 0.441 | | 7.270 | 0.000 |
| Logistics | 0.219 | 0.112 | 0.254 | 10.951 | 0.054 |
| Order picking | −0.023 | 0.118 | −0.024 | −0.194 | 0.846 |
| Product nature | 0.150 | 0.119 | 0.158 | 1.266 | 0.209 |
| Sustainability | −0.223 | 0.128 | −0.260 | −1.739 | 0.085 |
| Information sharing | 0.099 | 0.144 | 0.108 | 0.685 | 0.495 |
| Supply chain strategies | 0.048 | 0.155 | 0.054 | 0.311 | 0.757 |
| Logistics (Likely) | −0.009 | 0.108 | −0.012 | −0.085 | 0.932 |
| Order picking (likely) | −0.178 | 0.122 | −0.184 | −1.458 | 0.148 |
| Product nature (likely) | −0.030 | 0.115 | −0.037 | −0.259 | 0.796 |
| Sustainability (likely) | 0.092 | 0.113 | 0.107 | 0.821 | 0.414 |
| Information sharing (likely) | −0.184 | 0.137 | −0.194 | −1.340 | 0.184 |
| Supply chain (likely) | 0.115 | 0.127 | 0.136 | 0.907 | 0.367 |

Note: Coefficients[a] means Coefficient$^{0.05\%}$.

**Table A8.** Model significance for the regression on failure and cost increase control.

| | Coefficients[a] | | | | | |
|---|---|---|---|---|---|---|
| **Model** | **Unstandardized Coefficients** | | **Standardized Coefficients** | **t** | **Sig.** | |
| | **B** | **Std. Error** | **Beta** | | | |
| (Constant) | 3.052 | 0.436 | | 7.000 | 0.000 | |
| Logistics | 0.117 | 0.104 | 0.146 | 1.126 | 0.263 | |
| Order picking | 0.046 | 0.113 | 0.053 | 0.410 | 0.683 | |
| Product nature | 0.150 | 0.113 | 0.166 | 1.332 | 0.186 | |
| Sustainability | −0.108 | 0.125 | −0.133 | −0.864 | 0.390 | |
| Information sharing | −0.088 | 0.141 | −0.103 | −0.625 | 0.534 | |
| Supply chain strategies | 0.075 | 0.140 | 0.091 | 0.536 | 0.593 | |
| Logistics (Likely) | −0.017 | 0.098 | −0.023 | −0.175 | 0.862 | |
| Order picking (likely) | −0.113 | 0.115 | −0.122 | −0.979 | 0.330 | |
| Product nature (likely) | 0.077 | 0.102 | 0.102 | 0.753 | 0.453 | |
| Sustainability (likely) | 0.021 | 0.109 | 0.026 | 0.193 | 0.847 | |
| Information sharing (likely) | −0.042 | 0.132 | −0.047 | −0.318 | 0.751 | |
| Supply chain (likely) | 0.016 | 0.120 | 0.020 | 0.135 | 0.893 | |

Note: Coefficients[a] means Coefficient$^{0.05\%}$.

**Table A9.** Model significance for the regression on failure and service level.

| | Coefficients[a] | | | | | |
|---|---|---|---|---|---|---|
| **Model** | **Unstandardized Coefficients** | | **Standardized Coefficients** | **t** | **Sig.** | |
| | **B** | **Std. Error** | **Beta** | | | |
| (Constant) | 3.230 | 0.439 | | 7.354 | 0.000 | |
| Logistics | 0.019 | 0.105 | 0.024 | 0.186 | 0.853 | |
| Order picking | −0.157 | 0.115 | −0.175 | −1.367 | 0.175 | |
| Product nature | 0.105 | 0.112 | 0.116 | 0.930 | 0.355 | |
| Sustainability | −0.063 | 0.126 | −0.075 | −0.499 | 0.619 | |
| Information sharing | 0.076 | 0.141 | 0.085 | 0.539 | 0.591 | |
| Supply chain strategies | 0.147 | 0.141 | 0.173 | 1.043 | 0.300 | |
| Logistics (Likely) | 0.024 | 0.095 | 0.031 | 0.246 | 0.806 | |
| Order picking (likely) | −0.064 | 0.116 | −0.068 | −0.548 | 0.585 | |
| Product nature (likely) | 0.005 | 0.103 | 0.007 | 0.051 | 0.960 | |
| Sustainability (likely) | 0.033 | 0.112 | 0.039 | 0.294 | 0.769 | |
| Information sharing (likely) | −0.134 | 0.133 | −0.147 | −1.007 | 0.317 | |
| Supply chain (likely) | 0.130 | 0.120 | 0.158 | 1.079 | 0.283 | |

Note: Coefficients[a] means Coefficient$^{0.05\%}$.

**Table A10.** Summary statistics of response variables.

| Summary Statistics | DTC[p] | DC[p] | R[p] | V[p] |
|---|---|---|---|---|
| **Mean** | 0.802 | 0.766 | 0.268 | 0.352 |
| **Std. dev.** | 0.301 | 0.194 | 0.279 | 0.269 |
| **Min.** | 0.000 | 0.000 | 0.000 | 0.000 |
| **Max.** | 1.000 | 1.000 | 1.000 | 0.750 |

**Table A11.** Average marginal effects estimated by fractional multinomial logit.

| | Response Variables | | | |
|---|---|---|---|---|
| | **DTC[p]** | **DC[p]** | **R[p]** | **V[p]** |
| **Ind. variable:** $d_{\mathbb{F}}$ | | | | |
| Coefficient | −0.0915** | 0.0852** | 0.0001** | 0.0062** |
| Std. error | 0.0134 | 0.0124 | 0.0000 | 0.0018 |
| *z*-value | −68.16 | 68.34 | 17.13 | 33.50 |
| **Ind. variable:** $d_{\mathbb{V}}$ | | | | |
| Coefficient | 0.0697** | −0.0068 | 0.0001 | −0.0629** |
| Std. error | 0.0066 | 0.0061 | 0 | 0.0015 |
| *z*-value | 10.44 | −1.11 | −2.61 | −39.7 |
| **Ind. variable:** $d_{\mathbb{R}}$ | | | | |
| Coefficient | −0.009 | 0.0155** | −0.0024** | −0.0041 |
| Std. error | 0.0062 | 0.0059 | 0.0001 | 0.0007 |
| *z*-value | −1.43 | 2.60 | −18.39 | −5.38 |
| **Ind. variable:** $f_{\mathbb{F}}$ | | | | |
| Coefficient | −0.0731** | 0.0688** | 0.0001** | 0.0043** |
| Std. error | 0.0009 | 0.0009 | 0.0000 | 0.0001 |
| *z*-value | −73.59 | 74.72 | 17.29 | 30.96 |
| **Ind. variable:** $\delta_{\mathbb{F}}$ | | | | |
| Coefficient | −0.0569** | 0.0535** | 0.0000** | 0.0033** |
| Std. error | 0.0856 | 0.0800 | 0.0004 | 0.0115 |
| *z*-value | −66.53 | 66.92 | 16.31 | 29.10 |
| **Ind. variable:** $\delta_{\mathbb{V}}$ | | | | |
| Coefficient | 0.0603** | −0.0100** | −0.0001 | −0.0502** |
| Std. error | 0.0365 | 0.0326 | 0.0001 | 0.0105 |
| *z*-value | 16.52 | −3.08 | −0.76 | −47.52 |
| **Ind. variable:** $\delta_{\mathbb{R}}$ | | | | |
| Coefficient | −0.0459 | 0.0787** | −0.0123** | −0.0205** |
| Std. error | 0.0314 | 0.0297 | 0.0006 | 0.0037 |
| *z*-value | −1.46 | 2.64 | −18.17 | −5.45 |
| **Ind. variable:** $k_{\mathbb{R}}$ | | | | |
| Coefficient | 0.0162 | −0.0187 | 0.0015** | 0.0011 |
| Std. error | 0.0302 | 0.0286 | 0.0001 | 0.0034 |
| *z*-value | 0.53 | −0.65 | 14.86 | 0.31 |
| **Goodness-of-fit statistics** | Wald $\chi2$ = 23,869 | | | |
| | Log pseudolikelihood = −3,064.97 | | | |
| | Prob > $\chi2$ = 0.000 | | | |

Note: *Statistical significance*: **$p < 0.01$; *$p < 0.05$.

**Table A12.** DTC-fulfillment: Order fulfillment and warehouse operating costs.

| **DTC Order Fulfillment Costs** | | **DTC Warehousing Costs** | | | | | |
|---|---|---|---|---|---|---|---|
| $d_{\mathbb{F}}$ | $(\frac{d_{\mathbb{F}}}{d_{\mathbb{D}}})$ | $\mathbb{F}$ | $\mathbb{D}$ | $\mathbb{F}$ | $\mathbb{D}$ | $\mathbb{F}$ | $\mathbb{D}$ |
| $ 5.25 | 0.75 | 100% | | 100% | | 75% | 25% |
| $ 5.29 | 0.85 | 100% | | 65% | 35% | | 100% |
| $ 6.65 | 0.95 | | 100% | | 100% | | 100% |

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
