# Peer review of "Creating Sustainable Order Fulfillment Processes through Managing the Risk: Evidence from the Disposable Products Industry"

_sustainability, doi:10.3390/su12072871_

Round 1

Reviewer 1 Report

This paper tries to identify the sustainability factor as one of the most influential factor to the success of OFP in the disposable products industry. Overall, the paper seems to fail to understandably deliver the core results of the research. The contents of abstract do not fit to the title of the paper. The uniqueness and benefits of the proposed OFP are hardly figured out compared to the current one as the title says. The compelling evidence of sustainable factor is confusing enough to be concluded. The mathematical framework for the retail distribution system also does not seem to be relevant to the objective of this study, especially in terms of sustainability.

Author Response

Dear editorial board;

Thank you so much for your valuable comments on our paper, the following changes are done on this manuscript, and all changes are (Yellow highlighted).

  1. as respected editorial board requested the title of the paper changed to "Creating Sustainable Order Fulfillment Processes Through Managing the Risk: Evidence from the Disposables Products Industry"
  2. For better fitting the abstract, main context, and title abstract part rewritten.
  3. All of the paper’s grammar, and spell check revised with related software.
  4. Considering the new revise in the paper we added relevant sources.
  5. For better understanding, we added (5.2.4 Order Fulfillment: Fractional Multinomial Logit Model) as a new subsection for describing the mathematical model and discuss math model results and all of the details related to the model reported.
  6. We revised the conclusion part and discuss completely on our results and reported all of the details about our results and model.
  7. At the end one of the editors asked about checking (Disposables Products Industry), I should mention this, this research can apply to any industry and in the following paper, we used (Disposables Products Industry).

I am looking forward to hearing from you.

Sincerely yours,

Mohammad Heydari.

Reviewer 2 Report

The most important thing is to improve writing. There are lots of typos.

For example, in the abstract, "is the most important factors" should be "is the most important factor".

For the paper's title: "Creating Sustainable Order Fulfillment Process Through Managing the Risk Evidence from the Disposables Products Industry" should be revised as  "Creating Sustainable Order Fulfillment Processes Through Managing the Risk: Evidence from the Disposables Products Industry".

Plus, the term "Disposables Products Industry" is a bit strange to me. Please check.

Research rigor: I wonder whether there are any robustness testing schemes regarding the research assumptions of the method employed. The current analysis is a bit too simple. Some more details regarding research rigor should be provided.

Author Response

(The authors gave the same response as above.)

Round 2

Reviewer 1 Report

The manuscript has been considerably improved according to the reviewers' report. Since the 'sustainability' concept employed in the manuscript might be confusing enough to catch the key idea of this study, it needs to be clarified in the introduction, as a minor revision without further review.

Author Response

Dear editorial board;

Thank you so much for your valuable comments on our paper, the following changes are done on this manuscript, and all changes are (Yellow highlighted).

  1. As respected editorial board requested for better fitting the (INTRODUCTION) part re-written again.
  2. All of the paper’s grammar, and spell check revised with related software.
  3. Considering the new revise in the paper we added relevant sources.

I am looking forward to hearing from you.

Sincerely yours,

Mohammad Heydari.

Reviewer 2 Report

I think the revised paper is acceptable for publication .Thanks.

Author Response

(The authors gave the same response as above.)
